# Pro-Resolving Lipid Mediators in the Pathophysiology of Asthma

**DOI:** 10.3390/medicina55060284

**Published:** 2019-06-18

**Authors:** Oxana Kytikova, Tatyana Novgorodtseva, Yulia Denisenko, Marina Antonyuk, Tatyana Gvozdenko

**Affiliations:** Vladivostok Branch of Federal State Budgetary Science Institution “Far Eastern Scientific Center of Physiology and Pathology of Respiration”—Institute of Medical Climatology and Rehabilitative Treatment, Russian Street 73-g, Vladivostok 690105, Russia; nauka@niivl.ru (T.N.); karaman@inbox.ru (Y.D.); antonyukm@mail.ru (M.A.); vfdnz@mail.ru (T.G.)

**Keywords:** asthma, pro-resolving lipid mediators, eicosanoids

## Abstract

Asthma is one of the most important medical and social problems of our time due to the prevalence and the complexity of its treatment. Chronic inflammation that is characteristic of asthma is accompanied by bronchial obstruction, which involves various lipid mediators produced from n-6 and n-3 polyunsaturated fatty acids (PUFAs). The review is devoted to modern ideas about the PUFA metabolites—eicosanoids (leukotrienes, prostaglandins, thromboxanes) and specialized pro-resolving lipid mediators (SPMs) maresins, lipoxins, resolvins, protectins. The latest advances in clinical lipidomics for identifying and disclosing the mechanism of synthesis and the biological action of SPMs have been given. The current views on the peculiarities of the inflammatory reaction in asthma and the role of highly specialized metabolites of arachidonic, eicosapentaenoic and docosahexaenoic acids in this process have been described. The possibility of using SPMs as therapeutic agents aimed at controlling the resolution of inflammation in asthma is discussed.

## 1. Introduction

The problem of pandemic spread and the complexity of the treatment of modern chronic diseases, including asthma [1], may be successfully solved due to the achievements of multidimensional biology, the directions of which are combined by the term “omics” (genomics, transcriptomics, proteomics, metabolomics) [2]. The omics-biology approach can be aimed at identifying genomic, protein and metabolic changes in asthma and will provide a deeper understanding of the complex pathophysiological mechanisms of the chronicity of this disease and identify promising areas for finding effective therapeutic strategies. Lipidomics is the most important branch of metabolomics that involves the identification and quantitative measurement of lipids of a biological object. The study of lipidome is considered as a logical continuation of modern cell biology, aimed at the sequential description (or cataloging) of all molecular components of a cell. Recent advances in clinical lipidomics indicate its importance for studying metabolic processes [3]. Progress in lipidomics is due to the development of new mass spectrometry analytical methods [4] widely used for profiling and evaluating lipids [5]. The fact that lipids contribute to a wide range of homeostatic processes and pathological states has been established because of the development of this research area [6]. It is well known that lipids undergo enzymatic and non-enzymatic transformations into lipid mediators, which are involved in the processes of initiation (pro-inflammatory lipid mediators) and resolution (pro-resolving lipid mediators) of acute inflammation [7] (Table 1). Therefore, mediator lipidomics is currently the most actively developing scientific field [8]. Fatty acids (FAs) and their metabolites play a decisive role in regulating persistence and the resolution of inflammation in the diseases of the bronchopulmonary system. The metabolites of n-3 and n-6 polyunsaturated fatty acids (PUFAs) are the key for these processes. Thus, research in the field of mediator lipidomics has shown that arachidonic (AK), docosahexaenoic (DHA) and eicosapentaenoic (EPA) acids can act as precursors for the production of pro-inflammatory lipid mediators (eicosanoids (oxylipins)) and endogenous specialized pro-resolving mediators (SPMs) involved in the resolution of inflammation, including resolvins (Rvs), lipoxins (LXs), protectins (PDs) and maresins (MaRs) [9,10]. The study of these locally acting lipid mediators provides fundamentally new opportunities for the development of an effective strategy for treating chronic inflammatory diseases by correcting the processes of resolving inflammation, rather than suppressing it at the stage of initiation and development [11,12,13]. Despite the fact that the potential role of long-chain n-6 and n-3 FAs in health and disease has been an area of intensive research for many years, interest in studying lipid metabolites, which play a significant role in the resolution of inflammation, is constantly growing [14,15,16]. The use of n-3 FAs and/or the use of stable synthetic analogues of SPMs may significantly increase the effectiveness of treatment of chronic inflammatory diseases, in particular asthma [15,17].

The aim of the review is to summarize modern scientific information about the biological effects of SPMs—Rvs, LXs, PDs and MaRs—and determine opportunities for their potential using in therapy of asthma.

## 2. Ways of Biosynthesis of Lipid Mediators and Their Biological Effects

Various lipid mediators are involved in the regulation of inflammation, in particular, eicosanoids (oxylipins). They are oxidation products of FAs—namely, AK (20:4n-6), EPA (20:5n-3) and DHA (22:6n-3) (Figure 1). There are three pathways of oxylipin biosynthesis. Thromboxanes (Txs) and prostaglandins (PGs) are synthesized via the cyclooxygenase reaction with the participation of the cyclooxygenase (COX). The lipoxygenase (LOX) catalyzes the lipoxygenase pathway of the leukotriene (LT) biosynthesis. Cytochrome P450 is responsible for the generation of hydroxyeicosatetraenoic (HETE), epoxyeicosatrienoic (EET) and hydroxyoctadecanoic acids (HODE) via the epoxygenase pathway. There are several groups of eicosanoids, including the prostanoid, LT and eoxin families. The prostanoid family includes: PGs (PG2, PGF2α, PGD2, 15d-PGJ2) affecting smooth muscle tone of the bronchi, urogenital, vascular systems, gastrointestinal tract; prostacyclin (PGI2), which is a subspecies of prostaglandins and has an inhibitory effect on platelet aggregation and the vasodilation effect; and Txs, which causes vasoconstriction of small vessels. The LT family members cause bronchial muscle contraction. LOX is the target of currently available therapeutics (Montelukast, Zafirlukast). The biological effects of the eoxin family members (A4, C4, D4, E4) are associated with the development of allergy [18]. Oxylipins serve as secondary messengers of hydrophilic hormones, control the contraction of smooth muscle tissue (blood vessels, bronchi, uterus), are involved in the release of intracellular synthesis products (hormones, mucoids), affect bone metabolism, peripheral nervous and immune system, movement and aggregation of cells (leukocytes and platelets) and also are effective ligands for pain receptors. Oxylipins act on the cells synthesizing them (autocrine action) and on neighboring cells (paracrine action) by binding to membrane receptors as local bioregulators. The eicosanoids initiate the acute inflammatory process that is necessary for activating immune cells and synthesizing pro-inflammatory messengers. AK derivatives, namely, cysteine-containing LT and PGD2, play an important role in asthma pathogenesis. They are pro-inflammatory mediators, powerful bronchoconstrictors, cause hyperreactivity and swelling of the bronchi.

As soon as the number of polymorphonuclear leukocytes in inflammatory tissue decreases, the level of pro-inflammatory cytokines decreases and PUFA metabolism changes. The synthesis of pro-inflammatory lipid mediators (LTs and PGs) are switched to the production of SPMs (LXs, Rvs, PDs, MaRs), which have an anti-inflammatory and cytoprotective effect, affecting the resolution of the inflammatory reaction through the involvement of the immune system [14,19,20]. Switching the synthesis of pro-inflammatory lipid mediators (PG, LT) to the formation of resolving lipid metabolites (Rv, LX, PD, MaR) is considered as one of the key processes in the resolution of acute inflammation [9,12,13].

Therefore, the balance between n-6 and n-3 FAs and their metabolic products is closely interrelated with the pathogenesis of diseases, including chronic respiratory diseases [21,22,23,24]. A number of studies have shown that the change in composition and metabolism of FAs and the change in the profiles of lipid mediators in patients with asthma are accompanied by an imbalance of the regulatory mechanisms of the inflammatory process and contribute to its chronicity [23]. The clinical significance of PG and LT in asthma is well known, but the biochemical role of other lipid mediators in the regulation of airway tone and inflammation remains unclear [24]. Thus, the study of the role of endogenous SPMs in the resolution and the chronicity of inflammation in asthma [11,15,25,26] and their effect on the activity of neurogenic inflammation in this disease [27] are a hot area of research.

Maresins. It was discovered in 2007 that macrophages synthesize bioactive products with a pronounced anti-inflammatory effect, MaRs (macrophage mediator in resolving inflammation), from DHA [9]. MaR biosynthesis is mediated of 12-, 14-, 15-LOX. For example, there is a pathway of the DHA metabolism with the participation of 14-LOX, which generates a 7,14-dihydroxydoxahexaenoic acid called MaR1. Macrophages are key regulators of the inflammatory response. There are three subtypes of activated macrophages: classically activated (M1), alternatively activated (M2) macrophages and resolution-phase macrophages [28]. M1 macrophages are pro-inflammatory, M2 macrophages are associated with homeostasis restoration and tissue regeneration and the third subtype of macrophages has characteristics of M1 and M2 cells [29]. A number of studies have demonstrated that the number of M2 macrophages positively correlates with MaR1 level [9,28]. MaR1 inhibits the tissue infiltration by polymorphonuclear leukocytes and stimulates the phagocytic activity of macrophages. In vitro studies have shown that the direct impact of MaR1 on smooth muscle and endothelial cells results in the reduction of the production of pro-inflammatory cytokines and the decrease in the activation of nuclear factor kB (NF-κB) [26]. Recently, MaR2 that has a powerful bioregulatory effect was isolated [29,30].

Lipoxins. The substrate for LXs synthesis is AK. Two members of the LXs family, LXA4 and LXB4, have been well studied [31]. Besides that, it is known that epilipoxins are synthesized when taking aspirin (aspirin-triggered LX). LXs synthesis is catalyzed by 5-, 12-, 15-LOX. The precursor of these mediators is 15-HETE. In general, LXs are a branch of the leukotriene family. For example, their production by platelets is catalyzed by 12-LOX through converting LTA4 [32]. Unlike pro-inflammatory LTs, LXs act as powerful anti-inflammatory bioregulators, suppressing the inflammation and activating the processes of resolution and recovery [26]. The result of their action is the inhibition of chemotaxis and migration of macrophages and neutrophils to the inflammatory focus, blocking of lipid peroxidation, the activation of NF-kB and the suppression of the synthesis of pro-inflammatory cytokines. In addition, LXs are actively involved in functioning of macrophages that are associated with homeostasis restoration processes [32].

Resolvins. Rvs were identified in a mice model of inflammation in 2000 [7]. Rvs are synthesized by epithelial and endothelial cells and play a leading role among humoral factors contributing to the resolution of acute inflammation [33]. These mediators prevent the chemotaxis and the migration of macrophages and neutrophils to the inflammatory tissue, block intracellular signaling pathways and the synthesis of pro-inflammatory cytokines and chemokines, promote apoptosis of “spent cells”, stimulate the efferocytosis and the differentiation of macrophages towards M2 phenotype and regulate the functions of platelets. Rvs metabolism is mediated by 5-LOX, 15-LOX, cytochrome P450 and COX-2. Rvs of the E-series (RvE) are metabolites of EPA, Rvs of the D-series (RvD) are products of DHA. RvD epimers are formed from DHA during the use of aspirin. In addition, Rvs of the T-series (RvT), which are synthesized from DHA (RvT1, RvT2, RvT3, RvT4), have recently been found [34,35].

RvE1 and RvE2 are the main members of the E-series of the Rvs family. The anti-inflammatory action of RvE1 is due to the inhibition of the synthesis of cytokines, cell adhesion molecules, NF-kB expression, the chemotaxis of polymorphonuclear leukocytes and the migration of dendritic cells to the inflammatory focus. The biological effect of RvE2 is similar to RvE1. It regulates neutrophil chemotaxis, activates the phagocytosis and the synthesis of anti-inflammatory cytokines [36].

RvD are formed from DHA with the participation of 15- and 5-LOX. Currently, six types of RvD (RvD1, RvD2, RvD3, RvD4, RvD5, RvD6), which are synthesized from the intermediate product 17S-hydroperoxy-DHA, are known. RvD epimers are produced from DHA under the exposure to aspirin. The trivial names of RvD1 and its epimer formed in the presence of aspirin are 7S,8R,17S-trihydroxy-DHA and 7S,8R,17R-trihydroxy-DHA. RvD can stimulate the polarization of the pro-inflammatory macrophage phenotype towards the resolving M2-like phenotype [37], regulate the cytokine synthesis and inhibit the inflammation in experimental models of LPS-induced acute lung injury [38]. RvD1 is a regulator of the activity of polymorphonuclear leukocytes and is considered as a potential therapeutic agent for suppressing allergic reactions.

Protectins. PDs are formed from DHA with the participation of 15-LOX and are synthesized by a number of cells, including brain cells, monocytes and CD4^+^ T-lymphocytes [39]. PD1 (the key member of the PDs family) exhibits a strong anti-inflammatory and neuroprotective properties and has a pronounced ability to suppress the replication of the influenza virus [40]. Protectin D1 also inhibits the secretion of tumor necrosis factor α (TNF-α) and interferon γ (IFN-γ). Neuroprotectin D1 (PD1) firstly found in the nervous tissue limits damage in experimental cerebral ischemia by reducing the migration of polymorphonuclear leukocytes [35]. The action of this mediator is mediated by blocking the intracellular signaling pathways (NF-κB), reducing COX-2 expression and prostaglandin synthesis. PD1 is a regulator of the synthesis of proteins of the B-cell lymphoma 2 (BCL2) family, which have anti-apoptotic effect, and enhances the efferocytosis of apoptotic neutrophils [35].

## 3. The Receptors for SPMs

The action of endogenous SPMs is mediated via membrane LXA4 receptor/formylpeptide receptor 2 (ALX/FPR2) and cysteinyl leukotriene 1 receptor (CysLT1). In addition, there are G-protein-coupled receptor 32 (GPR32), chemokine receptor (CMKLR), LTB4 receptor 1 (BLT1) and unidentified surface receptors with high affinity for human polymorphonuclear leukocytes. The information about receptors for lipid mediators and their antagonists is presented in Table 2.

The ALX/FRP2 receptor belongs to GPRs and is present on neutrophils, eosinophils, monocytes, macrophages, fibroblasts and T-cells, as well as in the epithelium of the respiratory tract. The expression of LXA4 receptors (ALX) is regulated by inflammatory mediators, transcription factors and epigenetic mechanisms. The ALX receptor is involved in the transduction of LXA4, 15-epi-LXA4, RvD1 signals into a cell. In addition, LXA4 deficiency has been observed in severe asthma. It indicates an associative link between ALX signaling impairment and chronic lung inflammation [54].

Chemokine-like receptor 1 (CMKLR1), also known as Chemerin Receptor 23 (ChemR23) or RvE, belongs to GPRs and is expressed by brain cells, dendritic cells, epithelial cells and in kidneys. This receptor has also been found in the lungs and is involved in immunoregulatory mechanisms, as it plays an important role in the antiviral immunity.

GPR32 receptor, also known as RvD1 receptor (DDR1), is expressed on neutrophils, lymphocytes, macrophages and monocytes.

The newly discovered GPR18 receptor or RvD2 receptor (DRV2) is expressed on neutrophils, monocytes and macrophages [55]. RvD2 contributes to the resolution of inflammation through DRV2 receptor.

BLT1 receptor is expressed on human neutrophils, eosinophils, monocytes, macrophages, mast cells, dendritic cells and T cells.

Both BLT1 and ChemR23 are receptors for RvE1. RvE1 exhibits an antagonistic effect on BLT1 receptor by blocking the biological action of pro-inflammatory LTs. At the same time, RvE1 has a synergistic effect on ChemR23 by inhibiting the activation of NF-kB and enhances phagocytosis.

RvD1 acts via ALX and GPR 32 receptors. The ability of RvD5 interacts with GPR32 has also been demonstrated. A new receptor for RvD2 (GPR18) has been discovered. RvD1, RvD3 and RvD5 may bind to DRV1 receptor [55]. RvD1 interacts with ALX receptor at the resolution phase of inflammation. 

The receptors for PD1 and MaR1 has not been found yet [54].

Since the appearance of data on the immunological mechanism of action of SPMs and the discovery of a number of receptors, their interaction with transient receptor potential ion channels has become a matter of scientific interest. It is considered that chronic airway inflammation and mucus hypersecretion are associated with the sensitization to TRPV1. Nociceptors are involved in the pathogenesis of airway inflammation by initiating local neurogenic inflammation and activating the bronchoconstrictor mechanism [21]. It has been found that all SPMs negatively modulate the activity of ion channels [20]. It confirms the fact that they are not only potential analgesics, but can also decrease neurogenic inflammation in asthma [27]. It has been demonstrated that MaR1 negatively modulates the activity of TRPA1, blocking inflammation [9]. In addition, RvD1 and RvD2 have the ability to reduce the activity of ion channels, in particular TRPA1; RvD2 and RvE1 can inhibit the activity of TRPV1.

## 4. The Role of SPMs in the Pathogenesis of Asthma and Perspectives for Their Therapeutic Use

The impairment of SPM synthesis in asthma is an important pathogenetic mechanism of chronic inflammation and progression (Figure 2). MaR1 level firstly increases in response to lung damage compared to other SPMs [26]. This mediator, like all SPMs, regulates the resolution of inflammation by activating the efferocytosis of apoptotic neutrophils and tissue regeneration [9]. Recently, the asthma paradigm based on the role of the adaptive immune system has been significantly changed. It happened due to the establishment of the importance of type 2 innate lymphoid cells (ILC2) as an antigen-independent source of type 2 cytokines for the disease pathogenesis [56]. Krishnamoorthy et al. have shown that MaR1 plays an important role in the regulation of the functioning of ILC2 involved in asthma [55]. MaR1 increased de novo generation of regulatory T cells (Tregs), which interacted with ILC2 to suppress cytokine production in TGF-β-dependent manner. The exogenous administration of MaR1 (1 ng/mouse) to experimental animals during the allergic phase of asthma results in the decrease in eosinophils number in bronchoalveolar lavage, IgE, IL-5 and IL-13 levels and the increase in TGF-β concentration.

The incubation of MaR2 with isolated recombinant LTA4 hydrolase, which catalyzes the hydrolysis of LTA4 into the pro-inflammatory mediator LTB4, leads to the inhibition of LTB4 production. These data demonstrate not only the participation of the epoxy intermediate in the biosynthesis of maresins, but also their influence on the formation of the pro-inflammatory mediator LTB4.

The synthesis and biological activity of maresins produced by macrophages are initiated by 14-lipoxygenation of DHA with the formation of 14S-hydro (peroxy)-4Z, 7Z, 10Z, 12E, 14S, 16Z, 19Z-DHA and 13S, 14S-epoxy-4Z, 7Z, 9E, 11E, 13S, 14S, 16Z, 19Z-DHA (13S, 14S-eMaR), which catalyzed by 12-LOX. Then, this intermediate is enzymatically hydrolyzed to 7R, 14S-dihydroxy-4Z, 7R, 8E, 10E, 12Z, 14S, 16Z, 19Z-DHA (MaR1) or 13,14S-epoxy-4Z, 7Z, 9, 11, 13, 14S, 16Z, 19Z-DHA (MaR2) by epoxyhydrolase. 13S, 14S-eMaR is also a substrate for glutathione-S-transferase MU 4 (GSTM4) and LTC4 synthase (LTC4S). As the result of their interaction, MCTR1 (13R-glutathionyl, 14S-hydroxy-4Z, 7Z, 9E, 11E, 13R, 14S, 16Z, 19Z-DHA) is produced and then converted to MCTR2 (13R- cysteinyl glycine, 14S-hydroxy-4Z, 7Z, 9E, 11E, 13R, 14S, 16Z, 19Z-DHA) in participation of gamma-glutamyl transferase and MCTR3 (13R-cysteinyl, 14S-hydroxy-4Z, 7Z, 9E, 11E, 13R, 14S, 16Z, 19Z-DHA) by dipeptidase.

Elevated LTB4 level was observed in a number of inflammatory diseases, including asthma [17,23]. Therefore, it can be predicted that LTB4 production may be the potential therapeutic target for LT-mediated conditions, such as inflammation underlying asthma pathogenesis. It can be assumed that maresins may be the basis for a new promising therapeutic approach in asthma.

The impairment of production of LXA4 and LXB4 are associated with chronic inflammatory diseases, including asthma [57]. LXA4 is an endogenous mediator of mucosal inflammation and reduces the severity of allergic and asthmatic reactions. LXB4 is also expressed in mucosal tissues; however, its role in allergic inflammation is unknown. Karra et al. have demonstrated that LXB4 has an anti-inflammatory effect in mucosal inflammation of the upper and lower respiratory tracts in mice [58]. The pro-resolving action of this LX are determined by the ability of LXB4 to reduce the expression of specific receptors for IL-13, IL-13Rα1 and IL-13Rα2. LXA4 blocks the release of histamine from mast cells during their interaction with epithelial cells and reduces the neutrophilic degranulation of azurophilic granules. Besides the direct regulation of mast cells and eosinophils, LXB4 inhibits Th cell activation, pro-inflammatory cytokine release and neutrophil chemotaxis. The indirect mechanisms of the accelerated resolution of allergic lung inflammation are also possible by suppressing the activity of inflammatory mediators, such as granulocyte-macrophage colony-stimulating factor (GM-CSF). Together, these data highlight several cellular mechanisms of the regulation of allergic airway inflammation by LX. Low LXA4 level in exhaled breath condensate correlates with the deterioration of lung function [59]. Larsson et al. have showed differences in LX concentration in bronchial wash and bronchoalveolar lavage [60]. LX levels in bronchial wash were increased in asthma patients compared with the healthy group and there were no differences in LX levels in bronchoalveolar lavage between the groups. Differences in LX production in asthma may be important for the study of respiratory diseases. These mediators may be of considerable interest as a target for therapy of diseases of the broncho-pulmonary system.

It has been described in a number of reports that RvE1 has a powerful protective effect in airway inflammation [61], in particular in asthma [62]. The ability of RvE1 to reduce airway hyperresponsiveness has been shown in an asthma model in mice [63]. The anti-inflammatory properties of RvE1 are due to the decrease in cytokine mRNA in macrophages by suppressing the expression of the nuclear transcription factor NF-κB p65. The impact of RvD1 and aspirin-triggered RvD1 on the development of allergic airway reactions and their resolution was studied by Rogerio et al. [64]. They established that RvD1 and aspirin-triggered RvD1 are important modulators of allergic reactions of the respiratory tract, as they reduce the synthesis of pro-inflammatory mediators and eosinophilia. It has been found that RvD1 suppresses pro-inflammatory cytokines synthesis and airway hyperreactivity in mice with acute lung injury [65]. Mas et al. analyzed the concentration of RvD1 and RvD2 after n-3 PUFA taking [66]. It has been established that their levels were within the biological range in which their anti-inflammatory and pro-resolving activity is realized. Taking into account the key role of Rvs in the resolution of inflammation, the development of drugs based on derivates of these mediators, which will activate natural mechanisms of the resolution of inflammatory and intensify repair processes, may be a promising direction of modern research [9,35]. The influence of alimentary n-3 PUFAs on Rv profile was described in a number of studies [67,68]. D’Vaz et al. have indicated that n-3 PUFA administration reduces the incidence of infectious bronchopulmonary diseases [69]. The peroral administration of n-3 PUFAs combined with low doses of aspirin contributes to the conversion of EPA and DHA to RvD [9]. Recent studies have shown good prospects for using of Rvs in pulmonology [70].

The decrease in PD1 level is identified in severe and uncontrolled asthma [25]. PD1 has recently been revealed in exhaled air condensate in asthma exacerbations [71]. In addition, PD1 reduces the level of PGD2, a key prostanoid involved in airway hyperreactivity. The ability of PD1 to inhibit 15-LOX expression and, consequently, LT biosynthesis has been shown [71]. Intravenous injection of PD1 (2–200 ng) prior to administering an aerosol allergen, prevented the development of airway hyperresponsiveness, eosinophilic and T cell-mediated inflammation in allergen-sensitized mice. Intravenous injection of PD1 (20 ng) also accelerated the resolution of allergic airway inflammation [71]. The noted anti-inflammatory activity and anti-allergic effect of PD1, the successful results on the use of this lipid mediator in experimental respiratory diseases indicate a high therapeutic potential of PDs [72].

## 5. Conclusions

Asthma, as a chronic inflammatory disease of the respiratory tract, is one of the most important current health problems due to their prevalence and complexity of treatment. Modern achievements of molecular biology allow us to reveal many mechanisms of the inflammatory process in asthma. The basis of asthma pathogenesis is chronic airway inflammation that is mediated by various cells and mediators, as well as a change in their activity. Among the many biochemical mediators of inflammation, cysteine-containing LTs play the most important role in asthma. They are powerful bronchoconstrictors, increase vascular permeability and mucus secretion and directly affect the activation of eosinophils and the proliferation of bronchial smooth muscle cells. Eicosanoids contribute to the inflammatory process by triggering the acute inflammatory process, which is necessary for the expression of pro-inflammatory messengers and the activation of immune cells. The acute inflammatory reactions are protective, but without timely resolution they can lead to chronic inflammation. The rapid development of lipidomics allowed us to identify SPMs, new lipid derivatives of AK, EPA and DHA that are involved in the resolution of inflammatory process. The recently discovered biologically active substances include Rvs, LPs, PDs and MaRs. The disturbance of SPM synthesis is an important pathogenetic mechanism for the chronization of the inflammatory process in asthma and worsening the disease. The study of SPMs opens up fundamentally new possibilities for asthma therapy aimed at correcting the processes of the resolution of inflammation.

## Figures and Tables

**Figure 1 medicina-55-00284-f001:**
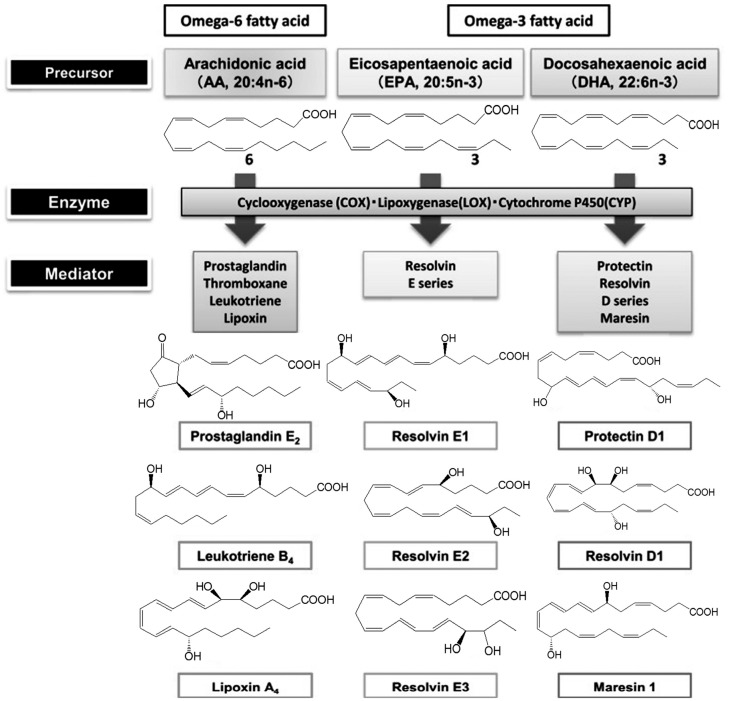
Synthesis of polyunsaturated fatty acid metabolites.

**Figure 2 medicina-55-00284-f002:**
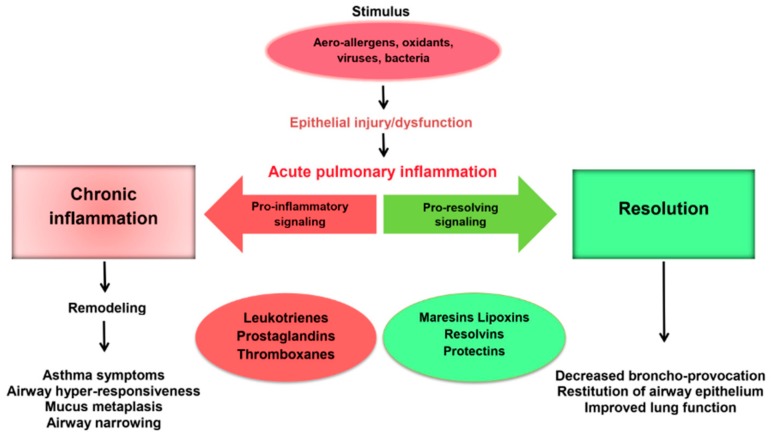
The role of lipid mediators in the development and resolution of inflammation in respiratory pathology.

**Table 1 medicina-55-00284-t001:** Pro-resolving and pro-inflammatory lipid mediators.

Pro-Resolving Lipid Mediators		Pro-Inflammatory Lipid Mediators	
MaRs	MaR1, MaR2	prostanoid family	PGE2, PG2, PGF2α, PGD2, 15d-PGJ2; PGI2; Txs
LXs	LXA4, LXB4		
Rvs	RvE (RvE1, RvE2), RvD (RvD1, RvD2, RvD3, RvD4, RvD5, RvD6), RvT (RvT1, RvT2, RvT3, RvT4)	leukotriene family	LTC4, LTD4, LTE4, LTF4, LTA4, LTB4
PDs	PD1	eoxin family	A4, C4, D4, E4

Resolvins (Rvs), lipoxins (LXs), protectins (PDs), maresins (MaRs), prostaglandins (PGs), thromboxanes (Txs), leukotriene (LT).

**Table 2 medicina-55-00284-t002:** Pro-inflammatory (PG, LT) and resolving lipid mediators (Rvs, LXs, PDs, MaRs) and their receptors.

Mediator		Receptor	Function	Antagonists Receptor	Literature
LT	LTC4	CysLT1 CysLT2	Bronchoconstriction	Leukotriene receptor antagonists: zafirlukast (akkolat, substance 1C1204219) pranlukast (substance ONO-1078), pobilukast (substance SKF 104353), montelukast (singular, substance ML-0476). Search for inhibitors of 5-lipoxygenase: zileuton (substance F-64077).	[41,42,43,44,45,46]
	LTD4	CysLT1 CysLT2	Bronchoconstriction		
	LTE4	CysLT1 CysLT2	Bronchoconstriction		
	LTF4	-	-		
	LTA4	-	-		
	LTB4	agonist for BLT1 BLT2	Mediates chemotaxis, plasma exudation, reduction of lung parenchyma		
PGs	PGI2	IP	Vasodilation, inhibitory effect on platelet aggregation	-	[47,48,49,50,51]
	PGE2	EP1	Bronchoconstriction	-	
		EP2	Bronchodilation	-	
		EP3	Activation of autonomic neurotransmitters	-	
		EPO	Pyregenic hyperalgesia	-	
	PGF2α	FP	Bronchoconstriction	-	
	PGD2	D-prostanoid (DP1), DP2 (CRTH2) and thromboxane prostanoid (TP)	Bronchoconstriction	TP antagonist: ramatroban or equivalent (TM30089). DP2 receptor antagonists: timipiprant (OC00459), BI 671800, setipiprant, MK-1029 and ADC-3680, feviprant	
MaRs	MaR1	-	-	-	-
	MaR2	-	-		
LXs	LXA4	agonist for ALX/FPR2; agonist for DRV/GPR32	Slowing chemotaxis and migration to the area of inflammation of macrophages and neutrophils, blocking lipid peroxidation, activation of NF-kB and inhibition of the synthesis of pro-inflammatory cytokines	-	[9,52]
Rvs	RvE1	antagonist for BLT1 and agonist for ERV/ChemR23	Decrease in airway Hyperresponsiveness, regulation of neutrophil chemotaxis, activation of phagocytosis and synthesis of anti-inflammatory cytokines	-	[9,36,53]
	RvD1	agonist for ALX/FPR2 agonist for DRV/GPR32	Reduced synthesis of proinflammatory mediators and eosinophilia	-	
PDs	PD1	-	Decreased airway hyperresponsiveness	-	[25,39,40]

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
