# Peer review of "Pro-Resolving Lipid Mediators in the Pathophysiology of Asthma"

_medicina, 2019, doi:10.3390/medicina55060284_

Reviewer 1 Report

The authors have introduced and reviewed pro-resolving lipid mediators in asthma.

The topic is interesting but there are too many details and appreciations in the paper that it is quite hard to read and becomes more like a list of different mediators. The major concerns about the review:

The paper should have at least 2 figures to get an overview of all the mediators. 

The target is not clear, sometimes the authors are talking about Th2-type asthma, but also aspirin-related asthma is covered. The asthma type should be defined at the beginning and then the story should be build around that. Or, if the idea is to cover certain type of mediators, review should be organised better based on that.

Tables are urgently needed. E.g. on the page 2, the introduction of different eicosanoids, prostanoids etc would work much better as a table, the same for maresins on page 5.

There is one reference for Figure on page 2 line 64, and one for Table on page 4 line 168, but at least I cannot see them anywhere.

Author Response

Dear reviewer,

We thank you for your careful consideration of our article. Your comments allowed improving the quality of the article. All corrections are highlighted in blue.

Point:  The target is not clear, sometimes the authors are talking about Th2-type asthma, but also aspirin-related asthma is covered. The asthma type should be defined at the beginning and then the story should be build around that. Or, if the idea is to cover certain type of mediators, review should be organised better based on that. Tables are urgently needed. E.g. on the page 2, the introduction of different eicosanoids, prostanoids etc would work much better as a table, the same for maresins on page 5. There is one reference for Figure on page 2 line 64, and one for Table on page 4 line 168, but at least I cannot see them anywhere.

Response: The main objective of the review was to provide general up-to-date information on the role of lipid mediators in the mechanisms of development and resolution of the inflammatory process in asthma, as well as to designate the prospects of using resolving mediators for different asthma phenotypes. Separation between asthma phenotypes was not conducted in this review. We added Table 1(Pro-resolving and pro-inflammatory lipid mediators) and Figure 2 (The role of lipid mediators in the development and resolution of inflammation in respiratory pathology).

We corrected:

Line 42 added Table 1

Line 65 added Figure 1

Line 173 added Table 2

Line 215 added Figure 2

Reviewer 2 Report

I read authors' review article "Pro-resolving lipid mediators in the pathophysiology of asthma" interestingly. Current knowledge about SPMs and its relationship with asthma was well described in this manuscript. So I think it can help the researchers of this field further.

Author Response

Dear reviewer,

We thank you for your careful consideration of our article. 

Reviewer 3 Report

This article contains lots of information with overview on the lipid mediators in the pathophysiology of asthma. However, I found it is hard to follow. I suggest author to provide a mechanistic diagram of how these lipid mediators contribute to the pathophysiology of asthma. It will help to comprehend these inter-correlated associations.  

Author Response

Dear reviewer,

We thank you for your careful consideration of our article. Your comments allowed improving the quality of the article. All corrections are highlighted in blue.

Point: This article contains lots of information with overview on the lipid mediators in the pathophysiology of asthma. However, I found it is hard to follow. I suggest author to provide a mechanistic diagram of how these lipid mediators contribute to the pathophysiology of asthma. It will help to comprehend these inter-correlated associations.

Response: We corrected:

Line 42 added Table 1

Line 215 added Figure 2

Reviewer 4 Report

In their interesting paper, Kytikova et al. describe in great detail the mechanisms of production and metabolism of different lipid metabolites derived from n-3 and n-6 PUFAs and their involvement in asthma. Taking into account the prevalence of asthma and the costs it involves, development of new, more effective therapies is of paramount importance. The article is well written and well-illustrated. There are few stylistic corrections, which I have included below. This topic is of great interest to the readership of the journal. I therefore find this article worthy of publication in “Medicina” pending this list of minor changes:

Line 13 “of” instead of “for” in “characteristic for asthma”

Line 22 “is being discussed” instead of “has being discussed”

Line 34 “as a logical continuation” instead of “as a logic continuation”

Line 45 “are the key for these processes” instead of “are a key for these processes”

Line 46 “has shown” instead of “have shown”

Line 73 “vasoconstriction” seems more appropriate than “narrowing” (which may sound permanent)

Line 73 – I suggest mentioning that lipoxygenase is the target of currently avialble therapeutics (Montelukast, zafirlukast)

Line 77 “are involved in the release” rather than “involve in the release”

Line 93-94 This statement is a little bit too far going, as there is a plethora of other processes taking place during the resolution of acute inflammation.

Line 97 – It is unclear to the reviewer what is meant by “the violation of composition”. Do you mean “change in composition”?

Line 103 – Unclear what is meant by “the topical highlights of medical research”; Do the authors mean something like “hot area of research”?

Line 143 – “RvE” – did the Authors meant RvE1?

Table 1 has the Russian words for “agonist for”, please replace:  “агонист для ALX/FPR2; агонист для DRV/ GPR32”

Line 213 – The meaning of “disease weighting” is unclear.

Line 217 – Replace “It has been happened” with “it happened”

Line 257 “colony” not “colone”

Author Response

Dear reviewer,

We thank you for your careful consideration of our article. Your comments allowed improving the quality of the article. All corrections are highlighted in blue.

Point:  

Line 12 “of” instead of “for” in “characteristic for asthma”

Line 21 “is being discussed” instead of “has being discussed”

Line 34 “as a logical continuation” instead of “as a logic continuation”

Line 46 “are the key for these processes” instead of “are a key for these processes”

Line 46 “has shown” instead of “have shown”

Line 74 “vasoconstriction” seems more appropriate than “narrowing” (which may sound permanent)

Line 75 – I suggest mentioning that lipoxygenase is the target of currently avialble therapeutics (Montelukast, zafirlukast)

Line 78 “are involved in the release” rather than “involve in the release”

Line 93-96 This statement is a little bit too far going, as there is a plethora of other processes taking place during the resolution of acute inflammation.

Line 99 – It is unclear to the reviewer what is meant by “the violation of composition”. Do you mean “change in composition”?

Line 105 – Unclear what is meant by “the topical highlights of medical research”; Do the authors mean something like “hot area of research”?

Line 145 – “RvE” – did the Authors meant RvE1?

Table 1 has the Russian words for “agonist for”, please replace:  “агонист для ALX/FPR2; агонист для DRV/ GPR32”

Line 213 – The meaning of “disease weighting” is unclear.

Line 219 – Replace “It has been happened” with “it happened”

Line 257 “colony” not “colone”

Response: We corrected:

Line 12 “of” instead of “for” in “characteristic for asthma”

Line 21 “is being discussed” instead of “has being discussed”

Line 34 “as a logical continuation” instead of “as a logic continuation”

Line 46 “are the key for these processes” instead of “are a key for these processes”

Line 46 “has shown” instead of “have shown”

Line 74 “vasoconstriction” instead of “narrowing”

Line 75 mentioned that lipoxygenase is the target of currently avialble therapeutics (Montelukast, zafirlukast)

Line 78 “are involved in the release” instead of “involve in the release”

Line 93-96 Corrected in the text.

Line 99 “change in composition” instead of “the violation of composition”   

Line 105 “hot area of research” instead of “the topical highlights of medical research”

Line 145 “RvE1” instead of “RvE”

Table 2 “agonist for ALX/FPR2; agonist for DRV/ GPR32” instead of “агонист для ALX/FPR2; агонист для DRV/ GPR32”

Line 213  Corrected in the text

Line 219  “it happened” instead of “It has been happened”

Line 257 “colony” instead of “colone”

Round  2

Reviewer 1 Report

The new, two additional tables have made the manuscript much more readable and easier to follow.

Just please change table 1 -> Table 1.